# Corporate governance and reporting quality of accounts in China-listed firms. A moderating role of ownership pattern

**Han Sun** [ID]*

School of Accountancy, Henan Institute of Economics and Trade, Zhengzhou, Henan, China

* claire_s@foxmail.com

## Abstract

Financial reporting quality is critical for businesses, stakeholders, and government to ensure transparency and accountability. The purpose of this paper is to investigate the relationship between corporate governance, financial reporting quality, and ownership structure as a moderating factor for Chinese stock exchange-listed firms. Quantitative data of 550 listed firms from 2012 to 2022 are collected from the annual reports. For investigating the relationship between variables, panel data analysis with random and fixed effect models is used. Our results show that corporate governance's different attributes such as Auditor brand name, Existence of an audit committee, independent board, family ownership, and profitability have a significant negative impact on the audit report lag that decreases the lags and increases the financial reporting quality in China listed firms. Auditor opinion, Board diligence Board size, and CEO duality have a significant positive impact on the audit report lag that increases the lags and decreases the financial reporting quality of China-listed firms. Furthermore, our findings show that ownership concentration has no moderating effect between corporate governance, different attributes, and financial reporting quality. Family ownership, on the other hand, has a strong moderating effect between corporate governance characteristics and financial reporting quality. However, due to limitations, this study provides the opportunity for future research on corporate governance mechanisms in different cultures and environments. Moreover, this study has some important implications for investors, policymakers, and government.

## Introduction

Corporate governance is an essential component of a healthy and sustainable business environment, and it has become increasingly important on a global scale [1]. Corporate governance provides the guidelines for the organization to set their rules and manage their operations within the organization. Corporate governance also provides polices to maintain accountability and transparency of their financial accounts. Corporate governance practices vary widely around the world, reflecting differences in legal systems, cultural norms, and business practices [2]. In some countries, corporate governance has been a top priority for decades,

**Data Availability Statement:** All relevant data are within the paper and its Supporting Information files.

**Funding:** This paper is a general project of 2021 Henan Higher Education Teaching Reform

Research and Practice Project (Research and Practice of "Integration of Competition and Teaching" in accounting major under the background of National Vocational College Skills Competition 2021SJGLX826) The funder wants to promote the higher education teaching in China and the funder has no role in study design, data collection, and analysis but provides the opportunities to the researchers to explore the problem that should be valuable in education reform.

**Competing interests:** The authors have declared that no competing interests exist.

while in others, it has only recently become a focus. One key development in corporate governance has been the emergence of global standards and best practices. In addition to the Global Governance Principles of the International Corporate Governance Network, these standards also incorporate the OECD Principles of Corporate Governance. The objective of these global standards is to establish a common understanding of what effective corporate governance entails and to provide recommendations to companies on how to achieve it [3].

China is selected in this study on the grounds of the frequent change of the code of corporate governance mechanism and also changes in the ownership pattern or shareholding pattern due to the rise in the foreign direct investment trend in the country. Furthermore, as China's economy has matured, there has been a shift in the types of FDI inflows. Previously, much of the investment was directed toward manufacturing and export-oriented industries. In recent years, there has been a greater emphasis on services, technology, and innovation-driven sectors. According to the [4], China enjoyed a significant increase in foreign direct investment (FDI) inflows. FDI into China increased by 5.74% in 2020, hitting USD 149.34 billion, up from USD 141.22 billion in 2019. Furthermore, the trend continued in 2021, reaching a new high of USD 180.95 billion. Corporate governance in China has undergone significant changes in recent years as the country has undergone rapid economic development and opened up to foreign investment [5]. The Chinese government has recognized the importance of strong corporate governance in maintaining the country's economic growth and has taken steps to improve corporate governance practices. One of the key changes in corporate governance in China has been the introduction of the Securities Law in 2019, which aims to improve the transparency and accountability of publicly traded companies. The law requires companies to disclose more information about their financial performance, corporate governance practices, and major transactions, and it imposes stricter penalties for violations. The China security exchange commission regulating and supervising China's securities market is another significant development [6]. The CSRC has introduced several rules and guidelines to improve corporate governance practices in China, including rules on independent directors, board composition, and shareholder rights. However, despite these improvements, corporate governance in China still faces several challenges. One of the biggest challenges is the dominance of state-owned enterprises (SOEs) in the economy, which can create conflicts of interest and make it difficult for minority shareholders to have their voices heard. Another challenge is the lack of independent directors and the prevalence of related-party transactions, which can raise concerns about conflicts of interest and undermine the trust of investors. Moreover, while corporate governance in China has improved in recent years, there is still room for further progress. Continued efforts to strengthen transparency, accountability, and the rights of minority shareholders will be essential for the long-term health and stability of China's economy [7].

Investor confidence in a business is largely dependent on having access to timely and accurate financial data. In developing financial markets, there is only source of financial information that is reliable in the capital market is the financial statement. Nonetheless, there is frequently a lag time between the conclusion of the audit and the release of the audited financial reports, which can reduce the efficiency of the capital markets. Before drafting legislation to shorten delays, authorities must comprehend the causes of this gap [8].

The main objective of this research is to assess the impact of organizational characteristics and corporate governance practices on the accuracy of financial reporting for companies listed in China. The study will specifically examine how ownership concentration and inter-organizational corporate governance (including the board of directors, audit committee, and CEO duality) are related to the reliability of financial reporting. Along with the primary goal, the study has several other specific objectives. Firstly, it aims to evaluate the current state of financial reporting quality in China. Secondly, the study intends to investigate the link between

corporate governance characteristics (such as board independence, size, CEO duality, and diligence) and financial reporting quality. Third, the study intends to investigate the relationship between auditor characteristics and financial report quality. Fourth, the study intends to look into the relationship between audit quality and financial report quality. The accuracy with which an audit reflects a company's financial performance and position is referred to as audit quality. Fifthly, the study aims to explore how ownership concentration moderates the relationship between corporate governance and the quality of financial reports for Chinese listed companies. Ownership concentration refers to the level of involvement and interest that company owners have in the governance and management of their companies and can include factors such as ownership size and concentration, the presence of institutional investors, and the level of shareholder activism. Lastly, the study aims to explore the moderating impact of family ownership on the relationship between corporate governance and the quality of financial reporting. The quality of financial reports will be measured by analyzing the audit reports from the annual financial reports of listed firms.

There are many studies [9, 10] conducted on the corporate governance of Asian economies and the developed economy but there is limited literature on the emerging Asian economy such as China that create interesting research gap for the further investigations. In the Asian economy there is state-owned and family-owned firms are dominant [11], and the investigations of family-owned firms are very interesting. Previous studies [12] are concerned with the linear relationship of corporate governance with financial reporting quality, we address this gap in the ownership structure such a family ownership and the ownership concentration as the moderating factors that may enhance the relationship between them. Moreover, there is little evidence of the panel data analysis by using random and fixed effect models in the previous literature, on that ground we found another interesting research gap for the investigation in our study. Moreover, the role of the auditor characteristics such as the auditors' opinions, auditor brand name and the auditor's independence from other services is important for the discussion that influences the financial reporting. However, another motive that motivate towards this study is the theoretical perspective, the previous literature is conducted on the developed economy, where the code of the corporate governance mechanism is consider strong and built their theories but in the Asian emerging economies situation is different.

The body of research pertaining to corporate governance and financial reporting quality is sustained by numerous theories, with some of the most widely recognized being agency theory, stewardship theory, and institutional theory. Agency theory posits that organizations are faced with a principal-agent predicament, wherein managers may prioritize their interests over those of shareholders. Effective corporate governance mechanisms, such as independent directors and effective monitoring, can assist in coordinating managers' interests with those of shareholders, leading to higher financial reporting quality [13]. Stewardship theory suggests that managers act as stewards of the organization and that they will act in the best interests of the organization if they are given the proper resources and support. Effective corporate governance mechanisms can provide the resources and support that managers need to act as effective stewards, which can lead to higher financial reporting quality [14]. The importance of social and cultural norms and values in shaping organizational behavior is emphasized in institutional theory. Effective corporate governance mechanisms can help organizations conform to societal expectations and norms, which can enhance their legitimacy and reputation [15]. This, in turn, can lead to higher financial reporting quality. Other theories that support the literature on corporate governance and financial reporting quality include signaling theory, stakeholder theory, and resource dependence theory [16]. Overall, these theories highlight the importance of effective corporate governance mechanisms in promoting high-quality financial reporting and ensuring the long-term success of organizations.

To fill the gap and obtain our objective, we used secondary data from the years 2012–2022 from the annual reports of the firms that are listed in China stock markets. The rationale behind selecting this time frame is that during this period, the corporate governance code in China underwent some modifications. Non-financial firms with the highest market capitalization are chosen at random as a sample from the total non-financial sector listed in China. For analysis, the panel data analysis with a random and fixed effect model is used. The accuracy of the model fitness is checked using the Housman specification test, which determines whether the model is fit and reliable.

Our results show that the Auditor brand name has a significant negative impact on the audit report lag that reduces the lags that increase the financial reporting quality. Auditor opinion has a positive impact on the audit report lag, indicating errors against the compliance and policy of the accounts then it takes time for an audit, also it means that the financial accounts have less quality and take too many errors to maintain. The existence of an audit committee reduces the lag and has complete control of the keeping of the accounts and has great internal control that increases the financial reporting quality in China. Board diligence has a significant and positive impact on the lags that reduce the account quality in China. The independent board decreased the lag and, in this way, financial reporting quality improved. In China, an increase in both board size and CEO presence can result in reporting delays and lower reporting quality. Family ownership, on the other hand, has a significant positive impact on financial reporting quality and can reduce audit reporting delays. It is also linked to increased profitability, decreased lag time, and improved financial reporting. It is worth noting that ownership density does not affect the relationship between corporate governance characteristics and financial reporting quality. On this issue, the regulatory impact of family ownership is significant.

This study makes significant contributions to the existing literature and theories on corporate governance and financial reporting quality. Firstly, it advances the development and refinement of established theories like agency theory, stewardship theory, and signaling theory. By exploring the connections between corporate governance mechanisms and financial reporting quality, the study identifies and elucidate the underlying processes and mechanisms that drive these relationships in various industrial and cross-cultural settings worldwide. Secondly, the study provides empirical evidence that demonstrates the impact of corporate governance on financial reporting quality. This evidence either supports or refutes existing theories and hypotheses. The study utilizes various empirical methods, including regression analysis and panel data analysis using random effects and fixed effect models. These methods help to establish a more comprehensive understanding of the complex relationships between corporate governance and financial reporting quality. Third, it provides great policy implications for regulators, standard-setters, and other stakeholders. By identifying the mechanisms and factors that influence financial reporting quality, this study helps to inform and guide policy decisions related to corporate governance, accounting standards, and financial reporting practices. Fourth, it contributes about the moderating impact of the family ownership and ownership concentration in the emerging markets, where the code of the corporate governance mechanism should consider weak as compared with the developed nation due to the structural pattern. Furthermore, this study help identify similarities and differences in corporate governance practices and their impact on financial reporting quality by examining the relationships between corporate governance and financial reporting quality in various countries and contexts.

The implications of studies on the impact of corporate governance on financial reporting quality are significant for companies, investors, regulators, and other stakeholders. The implications suggest that strong corporate governance is essential for ensuring the integrity of

financial reporting and protecting the interests of investors and other stakeholders. Companies that prioritize corporate governance, and investors and regulators that promote and enforce strong governance standards, are likely to benefit from higher financial reporting quality and lower risk of financial misreporting.

The remaining sections of this article are structured as follows: Part 2 presents an overview of recent literature and theoretical background and puts forward several hypotheses that can be tested to better understand the relationship between variables. The conceptual framework, research design, and methods are detailed in Part 3. Part 4 outlines the study's findings, while Part 5 encompasses the conclusion, suggestions for further research, limitations, and practical implications.

## Literature review

There are several theories that deal with the timing of financial reporting. In the literature, several theories have been put forth to clarify the connection between corporate governance and the timing delays in financial reporting. According to [17], one theory that supports this connection is agency theory. Effective corporate governance mechanisms play a critical role in mitigating agency conflicts. [18] noted that the application of agency theory within corporate governance can help to minimize such conflicts. [19] highlighted that corporate governance variables, such as ownership structure, board characteristics, and audit-related variables, can be instrumental in reducing agency problems within an organization. [20] further argued that good governance practices can limit agency conflicts and ensure the interests of shareholders are upheld. Financial managers also have a significant responsibility in mitigating conflicts by providing reliable financial information that benefits the organization while avoiding personal gain at the expense of shareholders.

Internal reporting theory posits that management is primarily focused on evaluating the company's internal performance [21]. In particular, managers tend to delay unfavorable news about the firm until it is confirmed as part of performance appraisal, which is tied to compensation and earnings performance. Consequently, managers may require additional time to prepare responses and address poor performance. [21, 22] also noted that management has a propensity to delay negative news. Conversely, good news is subject to less scrutiny and management tends to release it earlier than negative news.

IFRS emphasizes the importance of updating financial reports in a timely manner, as the information must be available to users when it is most useful in the decision-making process. The need for timely information underscores the requirement for external users to receive periodic financial reports for their evaluation in decision-making. Numerous studies have been conducted on various aspects of the timelines of financial reporting, with a focus on audit reports and financial report lags [20–22]. Despite the limited research in this area, several empirical studies have highlighted the significance of financial reports in the context of agency relationships. While financial reporting and institutional governance are closely related, they are frequently studied separately. The majority of these studies were carried out in developing countries. As a result, more research is required to investigate various financial reporting-related issues and to assess the impact of governance structures on financial report quality in emerging economies. One of the earliest studies to examine the timing of financial reports was conducted by [23] in the United States and investigated the determinants of annual report quality as measured by audit report delay and discovered that audit delay is related to strong audit opinion, poor internal control, and poor financial reporting quality.

Numerous studies conducted in developing countries have produced differing results regarding the relationship between the timely presentation of financial reports and financial

account quality. According to some studies, good news is broadcast before bad news [24], while others show the opposite [21]. According to some researchers, companies are hesitant to report bad news, so they use creative accounting techniques to avoid it. [25, 26]. A recent study in Malaysia looked at the relationship between corporate governance features (board independence, board expertise, and board rigidity) and audit report timing. The findings indicate a significant inverse relationship between the variables of the board of directors and audit report delay. According to the researchers, more frequent board meetings can help reduce audit report lags. However, there was no evidence of management independence or financial expertise among board members in the study, and the audit report was still delayed.

A research study conducted in Malaysia by [27] examined 628 firms to explore the relationship between corporate governance mechanisms and financial reporting quality. In this study, the audit committee and board of directors served as proxies for corporate governance. The results indicated that companies with effective and efficient financial reporting processes, influenced by the audit committee and board of directors, were more likely to submit financial reports on time. The study also found that organizations with more frequent audit committee meetings and members were more likely to submit audit reports on time. However, the study did not establish a conclusive link between the knowledge and independence of the audit committee and the timely submission of financial reports that demonstrate the quality of an organization's financial accounts. [28] investigated the quality of financial reporting of the three economies, Bangladesh, Pakistan, and India, and found audit report lag of 162, 145, and 92 days respectively and also concluded that if financial accounts were audited by the big audit firms, then they take less time due to the expertise.

Recently, [21] conducted a study on the quality of financial reporting and corporate governance, which revealed diverse findings on the reporting quality. Additionally, [29] investigated the influence of board members on financial reporting quality and found a positive impact due to ownership concentration in the firm structure. Moreover, [30] discovered a significant correlation between ownership concentration and reporting quality.

From a theoretical framework perspective, Agency theory describes the relationship between managers and the shareholders. It is assumed that the two parties have a natural conflict of interest because agents may prioritize their interests over those of the principals. This is referred to as an agency conflict. Corporate governance refers to the rules established by the government to control institutions and promote organizations by providing a level playing field. It seeks to address agency issues by aligning agents' interests with those of principals. The extent to which financial reports provide reliable and relevant information to stakeholders is referred to as financial reporting quality. Agency theory is relevant to both corporate governance and financial reporting quality because it helps to explain why and how agency problems arise. Agency theory suggests that in corporate governance, the principals should design governance mechanisms that align the interests of agents with those of the principals. Executive compensation plans that tie executive pay to company performance, for example, can motivate managers to act in the best interests of shareholders. According to the agency theory, managers may be tempted to manipulate financial reports to their benefit in terms of financial reporting quality. As a result, it is critical for principals to monitor and regulate financial reporting in order to ensure its quality. The use of independent auditors, for example, can provide assurance that financial reports are reliable and accurate. In conclusion, agency theory provides a theoretical foundation for understanding agency issues in corporate governance and financial reporting quality. It recommends that principals create governance mechanisms and monitoring systems that align agents' interests with those of the principals in order to ensure good corporate governance and high-quality financial reporting.

Numerous studies have been carried out to investigate the correlation between ownership concentration and financial reporting quality in Pakistani companies. The objective of these studies was to evaluate the effect of internal corporate governance mechanisms on financial reporting timelines. According to [31], ownership concentration has a positive but minor impact on the quality of financial reporting. The study was conducted in Malaysia using data from 2009 to 2012 for Malaysian listed firms. [32] revealed that there is a noteworthy adverse correlation between ownership concentration and the quality of financial reporting in cases where ownership concentration is low. However, in situations where ownership concentration is high, the relationship shifts to a positive correlation with the quality of financial reporting. In contrast, [33] found that state-owned firms in European countries are less conservative and have lower financial quality than other non-state-owned firms. Their study used data from 2003 to 2010.

[34] suggests that ownership concentration is a factor in financial reporting delays, with companies having block ownership and complex transactions experiencing higher reporting delays. However, [35] argue that concentration of ownership has no significant relationship with audit report delay. Overall, the findings of the preceding studies indicate that concentrated ownership has a negative impact on governance mechanisms, whereas eco-dominant ownership has a positive impact.

Ownership structure is critical in company governance. It can influence board composition, CEO compensation, and strategic decision-making through moderating the interaction between management and shareholders [36]. Recent corporate governance research frequently investigates how alternative ownership structures influence governance procedures and outcomes [37]. The relationship between ownership concentration and business performance moderated by ownership structure [38]. The impact of large institutional shareholders or family ownership, for example, on a company's performance might vary depending on the exact ownership structure [38]. According to [39] that a company's financial decisions, such as capital structure and dividend policy, can be influenced by its ownership structure. Different sorts of shareholders may have different risk, growth, or return preferences depending on ownership structure, financial decisions and outcomes. The ownership structure of a corporation influences the stakeholders, customers, employees, suppliers, and the community and also affects the stakeholder engagement and corporate social responsibility activities [40].

Family ownership has an impact on governance structures and the level of control inside a company [41]. Numerous studies such as [42] stated that family ownership influences corporate governance measures such as board composition, executive compensation, and minority shareholder rights protection. Therefore, these studies provide the gap in the literature to investigate the family ownership as moderating effect between corporate governance practices and the financial reporting quality. Moreover, according to [21] that ownership concentration frequently has an impact on corporate governance procedures. Recent research may dive into how concentrated ownership arrangements might either assist or hinder managerial control and monitoring. A dominant shareholder may be able to exercise significant influence on board composition, executive compensation, and strategic decisions due to high ownership concentration, potentially modifying corporate governance practices [43]. However, the degree of shareholder activism inside a corporation is influenced by ownership concentration [44]. On the grounds of the previous theoretical backgrounds and previous literature, we developed the following research hypothesis that is being tested to meet the research objectives.

**H1:** Board characteristics of corporate governance have a positive relationship with reporting quality of financials.

**H2:** Auditor characteristics of corporate governance have a positive relationship with reporting quality of financials.

**H3:** Ownership structure has a positive relationship with financial reporting quality.

**H4**: The firm's size and profitability have a positive relationship with financial reporting quality.

**H5:** Ownership Structure in the China-listed firms has a moderating impact between corporate governance attributes and financial reporting quality.

## Material and methods

### Data and sample size

In this study, we collected secondary data from the years 2012–2022 from the annual reports of the firms that are listed in China stock markets. The annual reports of selected firms are downloaded from the Shanghai Stock Exchange (SSE) data portal. The rationale behind selecting this time frame is that during this period, the corporate governance code in China underwent some modifications. The focus of this study is limited to the service and industrial sectors only, while the financial sector is excluded because the financial sector in China is governed by two regulators such as the Central Bank of China and the Securities Exchange Commission. Due to the dual monitoring by these organizations, the financial sector in China strongly follows the code of conduct, hence on that ground, we excluded the financial sector. If we consider the financial sector in our analysis, then our results will be biased. In this way, our study has limitations and advised future research suggestions to the researchers to take separate analyses and make comparisons between these sectors.

A total of 550 non-financial firms are selected as a sample from the total non-financial sector listed in China. There are 4697 listed firms belonging to the non-financial sector of China that represent our populations for the study. The Sample of 550 firms is collected through random sampling from non-financial sectors of the China-listed firms [21, 45, 46].

The conceptual framework is very useful for providing directions for the variables extracted from the previous theoretical and empirical background and provides the guidelines for the researchers to develop their study hypothesis on the ground. Fig 1 represents the conceptual framework of this study that developed on the grounds of the previous theory and empirical literature evidence. Moreover, Table 1 represents the variable's definitions and the measurement with the empirical evidence and references.

### Variables measurement and operational definitions

The audit report lag is the number of the days delay the audit report by the auditor to complete their audit report due to the numbers of the errors and non-compliance with the code of the corporate governance mechanism [47]. The total member in the board composition is an essential part of the corporate governance mechanism. The board size represents the number of the directors in making board decisions and taking necessary actions [48]. The independent board represents when there is independent directors involve in the board composition [48]. The CEO duality means that when CEO has the double role as CEO and the Chairman of the board of directors [50]. The numbers of the meeting of the board of directors held during the financial year is the board diligence [52]. The audit committee control and monitor the audit process within organization [54]. Moreover, the audit opinion may be qualified and unqualified, qualified means the auditor found irregularities their financial statement, otherwise

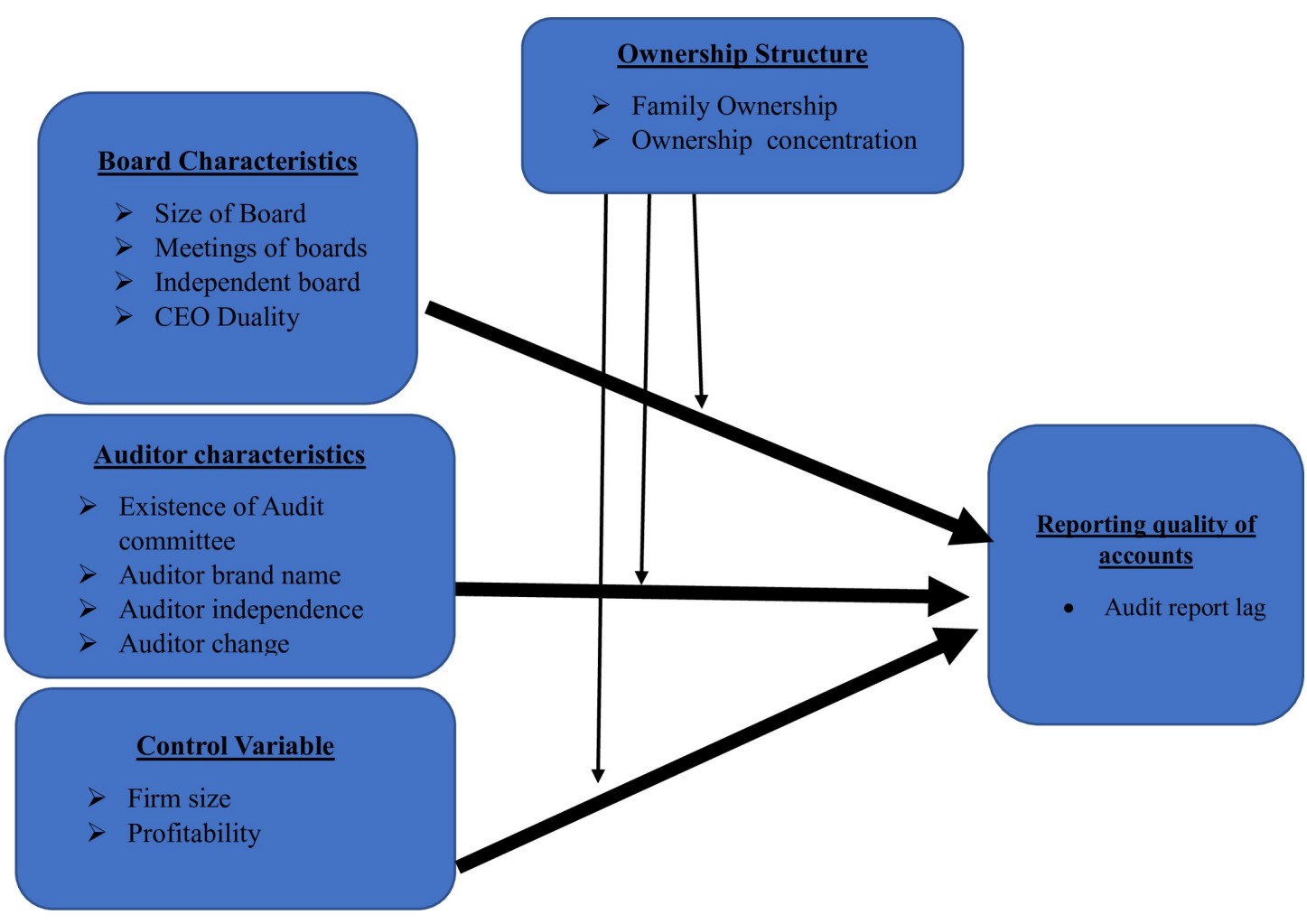

**Fig 1. Conceptual framework.**

**Table 1. Measurement of variable.**

| Variable | Sign | Measurement | Reference |
|---|---|---|---|
| Audit Report Lag | ARL | Measured in number of the day from the financial year end to auditor sign the audit report after the audit. | [47] |
| Board size | BIND | Total board of directors in total board composition. | [48, 49] |
| Board independence | BSIZ | Measured by using the proportion of independent directors in board composition | [17, 48, 50] |
| CEO duality | CEO | Combined role of the CEO then by dummy variable 1, otherwise 0 | [17, 51] |
| Board diligence | BDILIG | Measured the total number of board meetings in one financial period | [52, 53] |
| Audit committee | ACM | Presence of an audit committee, if present then by dummy variable 1 otherwise 0 | [17, 54] |
| Auditor opinion | AOP | Measured by if unqualified then dummy 1 otherwise 0 | [14] |
| Auditor independence | AOI | Auditor independence, measured by 1 if the auditor provides NAS, 0 otherwise | [55] |
| Auditor brand name | ABN | Auditor brand name used as a dummy variable 1 if the audit was conducted by big 4 audit firms otherwise 0. | [56] |
| Auditor change | ACH | Auditor change measured by dummy variable, denoted 1 if there is audit firm change, 0 otherwise | [57] |
| Ownership concentration | OCN | 5% or more percentage are held by a number of persons, institutes, or companies. | [58] |
| Firm Size | SIZE | Measured by the total assets of the organization as a proxy | [59] |
| Profitability | Pro | Profitably is measured by Earning per share | [59] |
| Family ownership | FOW | One member or family holds ownership measured by a major (more than 50%) share | [60] |

unqualified represents the compliance with code of the corporate governance mechanism by the organization [14]. Auditor brand name means the well reputed audit firms that conduct the audit of the listed firms authorized by the commission. Moreover, the auditor change is the change of the existing audit firm in the current year but most of the firms have different audit changes pattern. Ownership concentration means the shareholders in the shareholding pattern of the organization that hold 5% or more share percentage in the organization [58]. The family ownership means the shareholding by one member or family has 50% or more in the respect organization [60].

## Econometric model of the study

Model 1.

$$FRQ_{it} = \alpha_0 + \beta_1 BIND_{it} + \beta_2 BSIZ_{it} + \beta_3 CEO_{it} + \beta_4 BDILIG_{it} + \beta_5 ACM_{it} + \beta_6 AOP_{it}$$
$$+ \beta_7 ACH_{it} + \beta_8 ABN_{it} + \beta_9 AIND_{it} + \beta_{10} OCN_{it} + \beta_{11} PROFIT_{it} + \beta_{12} SIZE_{it} + \varepsilon_{it}$$

Model 2. Moderation Ownership concentration

$$FRQ_{it} = \alpha_0 + \beta_1 BIND_{it} + \beta_2 BSIZ_{it} + \beta_3 CEO_{it} + \beta_4 BDILIG_{it} + \beta_5 ACM_{it} + \beta_6 AOP_{it}$$
$$+ \beta_7 ACH_{it} + \beta_8 ABN_{it} + \beta_9 AIND_{it} + \beta_{10} PROFIT_{it} + \beta_{11} SIZE_{it} + \beta_{12} BIND*OCN_{it} + \beta_{13} BSIZ*OCN_{it}$$
$$+ \beta_{14} CEO*OCN_{it} + \beta_{15} BDILIG*OCN_{it} + \beta_{16} ACM*OCN_{it} + \beta_{17} AOP*OCN_{it} + \beta_{18} ACH*OCN_{it} +$$
$$\beta_{19} ABN*OCN_{it} + \beta_{20} AIND*OCN_{it} + \varepsilon_{it}$$

Model 3. Moderation Family ownership

$$FRQ_{it} = \alpha_0 + \beta_1 BIND_{it} + \beta_2 BSIZ_{it} + \beta_3 CEO_{it} + \beta_4 BDILIG_{it} + \beta_5 ACM_{it} + \beta_6 AOP_{it} + \beta_7 ACH_{it} +$$
$$\beta_8 ABN_{it} + \beta_9 AIND_{it} + \beta_{10} PROFIT_{it} + \beta_{11} SIZE_{it} + \beta_{12} BIND*FOW_{it} + \beta_{13} BSIZ*FOW_{it} + \beta_{14} CEO*FOW_{it}$$
$$+ \beta_{15} BDILIG*FOW_{it} + \beta_{16} ACM*FOW_{it} + \beta_{17} AOP*FOW_{it} + \beta_{18} ACH*FOW_{it} +$$
$$\beta_{19} ABN*FOW_{it} + \beta_{20} AIND*FOW_{it} + \varepsilon_{it}$$

## Methods of analysis

Panel data analysis is a valuable tool in social science research, as it allows for the examination of changes in individual or group behavior over time. Panel data analysis is a widely used approach in various disciplines such as economics, political science, sociology, and psychology. Two commonly employed techniques in panel data analysis are the random and fixed effects model. The random effects model treats unobserved heterogeneity as a random variable that is included in the error term, while the fixed effects model controls for individual-specific characteristics that are constant over time. Both models have been widely used in previous studies. In economics, panel data analysis is frequently used to examine the impact of policy changes or economic shocks on individual behavior, and fixed effects models are used to control for individual-level characteristics that may confound the relationship between policy change and behavior. In political science, panel data analysis is often used to examine the impact of institutional design on political behavior, and fixed effects models are employed to control for the fact that individuals in different institutions may have different baseline levels of political participation or other relevant characteristics. Therefore, panel data analysis using both random effects and fixed effects models is a powerful tool for understanding changes in individual or group behavior over time and has been extensively utilized across many social science

disciplines. Panel data analysis with random and fixed effects models has been utilized by numerous studies to investigate the association between corporate governance and financial reporting quality. [61] uses panel data from Chinese listed firms. [62] uses a quasi-natural experiment in China to examine the impact of board characteristics on financial reporting quality. [63] uses panel data from Korean firms. These studies showcase the effectiveness of utilizing panel data analysis with random and fixed effects models to investigate the correlation between corporate governance and financial reporting quality. They offer valuable insights into how various governance attributes can impact the financial reporting of the accounts. Moreover, the regression method of estimation is used for the moderation of the family ownership and ownership concentration that is very useful techniques for investigating the moderating impact.

## Findings and discussions

In statistics and data analysis, descriptive statistics and correlation matrices are useful tools. Descriptive statistics summarize a dataset in a succinct and relevant manner. They comprise metrics like the mean, median, variance, and standard deviation, which aid in understanding your data's central tendency, variability, and distribution. These summary statistics make complex datasets easier to comprehend and interpret. Descriptive statistics are necessary for initial data examination in our study. They assist in identifying outliers, trends, and patterns in data, which is useful in the early phases of data analysis. When comparing different datasets, descriptive statistics are frequently utilized. The descriptive statistics of variables are the primary indications of the checking problems in the data set such as normality and stationarity. Moreover, all these statistical indicators in our descriptive analysis are favorable that is a good sign for further analysis. A correlation matrix depicts how several variables in a dataset are related to one another. It quantifies the intensity and direction of these correlations, assisting you in determining whether variables tend to move in tandem or in opposite directions. Multicollinearity (strong correlations among predictor variables) in regression analysis pose issues with model interpretation and prediction. A correlation matrix aids in the detection of multicollinearity, allowing you to make necessary changes to your model. That's why we used the correlation matrix for initial screening of the directions of variables and the problems that existed between them.

Table 2 is related to the description of the samples. The Audit report lag (ARL) has a mean value of 67 days with a maximum of 91 days and a minimum of 2 days. The minimum number of days represents the good quality of the financial reports because every compliance and policy is followed by the firm and takes less time. On this mentality audit report lag represents the financial reporting quality [64]. Similarly, about 80% of firms take their audit from the well-reputed 4 audit firms globally. About 95% of firms have the existence of an audit committee. There are about 95% have qualified audit reports from the selected sample firms in China. The board's meeting has an average of 5 meetings in a single financial year. The average board size has 9 members in total board composition and 13% of firms have CEO duality. In our data there is 23% are governed by the family in China which has less as compared with other Asian economies. In Table 3, there is not any evidence of the high correlation existing between all selected variables. All the values between the variable of the correlation is less than 0.80 which represents the accurate correlation between them. If values in the shape of negative or positive are greater than 0.80 then represents the high positive or negative correlation coefficient between the variables.

The correlations between the audit report lag and auditors' brand name are -0.2046 which shows the negative and week correlations between them. Moreover, audit lag also has negative

**Table 2. Sample data description.**

| Variable | Mean | Median | Maximum | Minimum | Std. Dev. |
|---|---|---|---|---|---|
| ARL | 67.442 | 67.5 | 91.169 | 2 | 31.9736 |
| ABN | 0.804 | 1 | 1 | 0 | 0.39737 |
| ACH | 0.014 | 0 | 1 | 0 | 0.11761 |
| ACM | 0.952 | 1 | 1 | 0 | 0.21398 |
| AIND | 0.2 | 0 | 1 | 0 | 0.4004 |
| AOP | 0.952 | 1 | 1 | 0 | 0.21398 |
| BDILIG | 5.426 | 5 | 24 | 1 | 2.22945 |
| BIND | 0.718 | 1 | 1 | 0 | 0.45042 |
| BSIZE | 9.04 | 9 | 17 | 7 | 2.04224 |
| CEO | 0.136 | 0 | 1 | 0 | 0.34313 |
| FOW | 0.23 | 1 | 1 | 0 | 0.3004 |
| LOGSIZE | 10.1483 | 10.0919 | 22.5601 | 6.744694 | 1.97134 |

weak correlations between auditor independence, auditor opinion, board independence, and family ownership are -0.068, and -0.339. -0.099 and -0.1068, respectively. The audit report lag has a positive weak correlation with auditor change, the existence of audit committees, board meetings, board size, and CEO duality. The Auditor's brand name has positive week correlations with auditor change, auditor independence, auditor opinion, board diligence, board

**Table 3. Correlation matrix.**

| | ARL | ABN | ACH | ACM | AIND | AOP |
|---|---|---|---|---|---|---|
| ARL | 1 | | | | | |
| ABN | -0.20469 | 1 | | | | |
| ACH | 0.011141 | 0.015952 | 1 | | | |
| ACM | 0.033863 | -0.04016 | 0.026756 | 1 | | |
| AIND | -0.06875 | 0.158703 | 0.153204 | -0.05146 | 1 | |
| AOP | -0.33901 | 0.12482 | 0.026756 | -0.05042 | 0.088882 | 1 |
| BDILIG | 0.252508 | 0.049196 | -0.02279 | 0.017744 | -0.02604 | -0.15449 |
| BIND | -0.09903 | 0.048862 | 0.036847 | 0.191955 | -0.09778 | -0.07835 |
| BSIZE | 0.026644 | 0.271443 | -0.07743 | -0.04604 | -0.08087 | -0.06897 |
| CEO | 0.019717 | -0.03927 | 0.052043 | 0.089087 | 0.093352 | -0.04738 |
| FOW | -0.10668 | 0.100582 | -0.12157 | -0.2201 | -0.15003 | 0.098071 |
| LOGSIZE | 0.02765 | -0.07711 | -0.03524 | 0.066683 | -0.09066 | 0.060648 |
| | BDILIG | BIND | BSIZE | CEO | FOW | LOGSIZE |
| ARL | | | | | | |
| ABN | | | | | | |
| ACH | | | | | | |
| ACM | | | | | | |
| AIND | | | | | | |
| AOP | | | | | | |
| BDILIG | 1 | | | | | |
| BIND | 0.022084 | 1 | | | | |
| BSIZE | 0.193874 | 0.108144 | 1 | | | |
| CEO | -0.12828 | 0.015248 | -0.18222 | 1 | | |
| FOW | -0.00832 | -0.07193 | -0.13178 | 0.248833 | 1 | |
| LOGSIZE | 0.119673 | 0.054344 | 0.087265 | 0.022551 | 0.190839 | 1 |

independence, board size, and family ownership, respectively. Moreover, the audit brand name has negative week correlations with the existence of audit committee and CEO duality. Auditor change has a positive weak correlation with all the variables except board diligence, the board size, log of firm size, and family ownership, with these it has a negative correlation. The audit committee has a negative week and positive both types of behavior in respect of the correlation with other variables, it has a positive week correlation with board diligence, board independence, and CEO duality, and other has a negative week correlation with the existence of audit committee. The auditor independence shows a negative weak correlation with all the variables except CEO duality. However, the auditor opinion shows weak correlations with all selected variables except family ownership and log of the size of firms, where it has a positive weak correlation. Board diligence has a positive weak correlation with all variables except CEO duality and family ownership, with these it shows a negative weak correlation. Family ownership has a positive week correlation value of 0.1980 with a log of the size of firms.

## Results

As discussed earlier, our data has the characteristics of the balanced panel. Panel data is analyzed through random and fixed effect models if there is a random or fixed effect found in the industrial setup. The effectiveness of the model is checked through the Houseman specification test (Table 4) to see which method is effective. In our study analysis, we run both models and check the reliability of the model. Then Fixed effect model results (Table 5) are more reliable than the random effect model results (Table 6) because the Houseman specification test is significant and suggests that Fixed is effective and reliable.

Fixed effect model results in Table 5 show that the Auditor brand name has a significant negative impact on the audit report lag. It means that the auditor brand name reduces the lags that increase the financial reporting quality. Auditor opinion if qualified has a positive impact on the audit report lag because the results are significant and positive, it means that if there are some errors against the compliance and policy of the accounts then it takes time for an audit, also it means that the financial accounts have less quality that takes too many errors in maintained. The existence of an audit committee reduces the lag; it means that the Audit Committee has complete control of the accounts keeping and has great internal control that increases the financial reporting quality in China. Board diligence has a significant and positive impact on the lags. It means that a large number of board meetings increase the lag and reduce the account quality in China due to the wastage of time. The independent board decreased the lag because our result is significant and negative, which means that when the lag is decreased then financial reporting quality improves. Board size and CEO have the burden for the firms because our findings show that the result is significant and positive, which means that both increase the lag and decrease the reporting quality in China. Family ownership has a significant negative impact on the audit report lag and increases the financial reporting quality because family involvement has much interest in the firm's operation and improves the accounts reporting quality. Moreover, profitability also decreases the lag and increases the financial reporting quality of accounts because companies have no need for window-dressing and reporting the original words in their accounts timely. In Table 7, our results find that no

**Table 4. Hausman specification test.**

|  | Chi-Statistic | Chi-Sq | P-Value. |
|---|---|---|---|
| Cross-section random | 24.163953 | 13 | 0.0297** |

(Significance level 1% represents by***, 5% represents ** and 10% represents *)

**Table 5. Fixed effect method results.**

| Variable | Coefficient | Std. Error | t-Statistic | Prob. |
|---|---|---|---|---|
| ABN | -16.4355 | 4.250417 | -3.86679 | 0.0001*** |
| ACH | 8.078762 | 12.25231 | 0.659367 | 0.5101 |
| AOP | -37.6067 | 6.847966 | -5.49167 | 0.0000*** |
| AIND | -3.60826 | 4.500675 | -0.80172 | 0.4232 |
| ACM | 7.479195 | 6.854457 | 1.091143 | 0.0059*** |
| BDILIG | 3.998567 | 0.671554 | 5.954205 | 0.0000*** |
| BIND | -11.4942 | 3.381665 | -3.39898 | 0.0007**** |
| BSIZ | 1.080284 | 0.77585 | 1.392388 | 0.1006* |
| CEO | 10.05664 | 4.63144 | 2.171385 | 0.0305** |
| FOW | -5.26728 | 3.545435 | -1.48565 | 0.1002** |
| OCN | 2.100229 | 1.69785 | 1.236994 | 0.2168 |
| FSIZ | 0.914416 | 0.758705 | 1.205232 | 0.2289 |
| PROFIT | -0.09001 | 0.050085 | -1.79716 | 0.0731** |
| R-squared | 0.064551 | | | |
| Adjusted R-squared | 0.062459 | | | |
| Prob | 0.00000 | | | |
| Durbin-Watson stat | 2.011305 | | | |

(Significance level 1% represents by***, 5% represents ** and 10% represents *)

moderating impact is found between corporate governance's different attributes and ownership concentration. There are different reasons behind it due to the cultural, social, and other attributes effect. In Table 8, our results show that family ownership has a strong moderating

**Table 6. Random effect results.**

| Variable | Coefficient | Std. Error | t-Statistic | Prob. |
|---|---|---|---|---|
| ABN | -12.2562 | 3.575693 | -3.42764 | 0.0007*** |
| ACH | 8.157118 | 11.13667 | 0.732455 | 0.4642 |
| AOP | 37.2669 | 6.125766 | -6.08363 | 0.0000*** |
| AIND | -3.43742 | 3.463896 | -0.99236 | 0.3215 |
| ACM | -6.708956 | 6.391417 | 1.049682 | 0.0944* |
| BDILIG | 3.240303 | 0.60236 | 5.379342 | 0.0000*** |
| BIND | -9.37136 | 2.948473 | -3.17838 | 0.0016*** |
| BSIZ | 0.640928 | 0.691188 | 0.927285 | 0.0542** |
| CEO | 5.426485 | 4.065259 | 1.334843 | 0.1006* |
| FOW | -4.4073 | 3.019509 | -1.45961 | 0.1005* |
| OCN | 0.729321 | 0.932808 | 0.781856 | 0.4347 |
| FSIZ | 0.688943 | 0.67784 | 1.01638 | 0.3111 |
| PROFIT | -0.08807 | 0.043281 | -2.03478 | 0.0424** |
| C | 84.52994 | 11.49178 | 7.355687 | 0 |
| R-squared | 0.08711 | | | |
| Adjusted R-squared | 0.07341 | | | |
| Prob | 0.00000 | | | |
| Durbin-Watson stat | 2.091305 | | | |

(Significance level 1% represents by***, 5% represents ** and 10% represents *)

**Table 7. Ownership concentration as moderating variable.**

| Variable | Coefficient | Std. Error | t-Statistic | Prob. |
|---|---|---|---|---|
| C | 71.35368 | 13.52362 | 5.276228 | 0 |
| ABN | -15.3049 | 8.322264 | -1.83903 | 0.0667* |
| ACH | 27.57847 | 28.88093 | 0.954902 | 0.3402 |
| AOP | -15.4517 | 13.02778 | -1.18605 | 0.2363 |
| AIND | 7.315001 | 9.114772 | 0.802543 | 0.4227 |
| ACM | -10.1634 | 12.03974 | -0.84415 | 0.3991 |
| BDILIG | -0.03488 | 1.612812 | -0.02163 | 0.9828 |
| BIND | -13.4392 | 7.946038 | -1.69131 | 0.0916* |
| BSIZ | 3.333755 | 1.500197 | 2.222212 | 0.0269** |
| CEO | 6.932311 | 9.733703 | 0.712197 | 0.4768 |
| FSIZ | 0.762956 | 0.749055 | 1.018558 | 0.3091 |
| PROFIT | -0.11547 | 0.047918 | -2.40977 | 0.0164** |
| ABN*OCN | -0.73069 | 2.658048 | -0.2749 | 0.7835 |
| ACH*OCN | -9.90507 | 13.04546 | -0.75927 | 0.4482 |
| AOP*OCN | -7.60236 | 4.089501 | -1.85899 | 0.0638 |
| AIND*OCN | -3.5084 | 3.104005 | -1.13028 | 0.2591 |
| ACM*OCN | 9.517761 | 4.700449 | 2.024862 | 0.0436 |
| BDILIG*OCN | 1.381227 | 0.540905 | 2.55355 | 0.0111** |
| BIND*OCN | 0.294313 | 2.673271 | 0.110095 | 0.9124 |
| BSIZ*OCN | -0.75481 | 0.473114 | -1.59541 | 0.1115 |
| CEO*OCN | 1.069511 | 3.241534 | 0.32994 | 0.7416 |
| R-squared | 0.19812 | | | |
| Adjusted R-squared | 0.17910 | | | |
| Prob | 0.00000 | | | |
| Durbin-Watson stat | 1.91167 | | | |

(Significance level 1% represents by***, 5% represents ** and 10% represents *)

impact between corporate governance attributes and financial reporting quality. In Asian economies, mostly family-owned firms perform well due to the dual internal control and pressure by the family on the corporation that reduces the lags and increases the financial reporting quality. Moreover, Table 9 explains the hypothesis summary of accepted and rejected with the appropriate reasons with the additions of the addition in the existing theory and empirical literature.

## Discussion on findings

Our results show that the Auditor brand name has a significant negative impact on the audit report lag. It means that the auditor brand name reduces the lags that increase the financial reporting quality. However, the relationship between the auditor brand name and the financial reporting quality is based on different factors that include the auditors' skills, the size of the company, and the bookkeeping according to IFRS or not, and so on. Thus, it is crucial to take into account the particular circumstances when evaluating the effect of auditor brand name on audit report lag and financial reporting quality. Furthermore, the relationship between auditor brand name, audit report lag, and financial reporting quality might differ across various countries and regions due to variations in legal, regulatory, and cultural aspects. However, our

**Table 8. Family ownership as moderating variable.**

| Variable | Coefficient | Std. Error | t-Statistic | Prob. |
|---|---|---|---|---|
| ABN | -12.2562 | 3.575693 | -3.42764 | 0.0007*** |
| ACH | 8.157118 | 11.13667 | 0.732455 | 0.4642 |
| AOP | -37.2669 | 6.125766 | -6.08363 | 0.0000*** |
| AIND | -3.43742 | 3.463896 | -0.99236 | 0.3215 |
| ACM | 6.708956 | 6.391417 | 1.049682 | 0.2944 |
| BDILIG | 3.240303 | 0.60236 | 5.379342 | 0.0000*** |
| BIND | -9.37136 | 2.948473 | -3.17838 | 0.0016*** |
| BSIZ | 0.640928 | 0.691188 | 0.927285 | 0.0542** |
| CEO | 5.426485 | 4.065259 | 1.334843 | 0.1006* |
| FOW | -4.4073 | 3.019509 | -1.45961 | 0.1005* |
| OCN | 0.729321 | 0.932808 | 0.781856 | 0.4347 |
| FSIZ | 0.688943 | 0.67784 | 1.01638 | 0.0031*** |
| PROFIT | -0.08807 | 0.043281 | -2.03478 | 0.0424** |
| ABN*FOW | -11.4565 | 3.7961 | -3.01796 | 0.0027*** |
| ACH*FOW | 11.48943 | 11.88698 | 0.966555 | 0.0342*** |
| AOP*FOW | -16.7413 | 5.855062 | -2.85928 | 0.0044*** |
| AIND*FOW | -0.64165 | 3.63609 | -0.17647 | 0.0006*** |
| ACM*FOW | 26.40588 | 6.236226 | 4.234272 | 0.0000*** |
| BDILIG*FOW | 3.930914 | 0.633287 | 6.207161 | 0.0000*** |
| BIND*FOW | -8.19827 | 3.140469 | -2.61052 | 0.0093**** |
| BSIZ*FOW | 2.672931 | 0.675743 | 3.955545 | 0.0001**** |
| CEO*FOW | 6.94671 | 4.325415 | 1.606022 | 0.1009* |
| FOW*FOW | -1.56455 | 3.189772 | -0.49049 | 0.0024*** |
| OCN*FOW | 1.610614 | 0.955601 | 1.685446 | 0.0925* |
| FSIZ*FOW | 2.3695 | 0.682479 | 3.4719 | 0.0006*** |
| PROFIT*FOW | -0.08556 | 0.046135 | -1.85462 | 0.0643* |
| R-squared | 0.45190 | | | |
| Adjusted R-squared | 0.50812 | | | |
| Prob | 0.00000 | | | |
| Durbin-Watson stat | 2.161043 | | | |

(Significance level 1% represents by***, 5% represents ** and 10% represents *)

results are similar to the previous findings of [65] that found larger audit firms, which are generally more well-known and reputable, have shorter audit report lags in China.

Auditor opinion if qualified has a positive impact on the audit report lag because the result is significant and positive, it means that if there are some errors against the compliance and policy of the accounts then it takes time for an audit, also it means that the financial accounts have less quality that takes too many errors in maintained. An auditor's qualified opinion means that there are material misstatements in the financial statements that have not been corrected or disclosed, which could have a negative impact on financial reporting quality. In this case, the audit may take longer to complete as the auditor needs to investigate and verify the nature and extent of the misstatements, which could lead to a longer audit report lag. Furthermore, a qualified opinion can have a negative impact on a company's reputation and may lead to increased scrutiny from regulators and investors. Therefore, it is generally in the company's best interest to address and correct any material misstatements in a timely manner to avoid a qualified opinion. Our findings are consistent with the previous findings [65] that found

**Table 9. Hypothesis decisions and reasons.**

| Hypothesis | Hypothesis statement | Decision | Reasons |
|---|---|---|---|
| *H1* | Board characteristics of corporate governance have significant positive relationship with reporting quality of financials. | Accepted | Board characteristics such as board diligence, board size, and CEO duality increase the audit lags leading to a decrease in the financial reporting quality of the organizations. The large board size and the dual role of CEO create the disputes in the decision-making, while large meetings of the boards show the dispersed brainstorming on the problems that lead to an increase in the lag. Independent directors increase financial reporting quality by taking decisions without any influence. |
| *H2* | Auditor characteristics of corporate governance have significant positive relationship with reporting quality of financials | Partially accepted | Our alternative hypothesis partially accepts this case. Only the audit brand name and the audit opinions are significant. Auditing from big and well-reputed audit firms leads to a decrease in the audit lag because they have an efficient team and take timely audits that lead to improving the financial reporting quality. The existence of the audit committee increases the quality by reducing the lags due to control over the financials. Auditor change and auditor independence are not significant because these factors do not matter in China due to cultural changes. |
| *H3* | Ownership structure has a significant positive relationship with financial reporting quality. | Partially accepted | Family ownership has control over the firms leads to a reduction in audit report lag and an increase in the reporting quality in China, while the ownership concentration has not been significant because in China due to the different aspects such as culture domain the shareholders of more than 5 percent share has no influence over the organizations. |
| *H4* | The firm's size and profitability have a significant positive relationship with financial reporting quality. | Partially accepted | Larger firms do have not a significant impact on financials reporting in China due to the strong code of mechanisms that prevails in the country, but profitability leads to an increase in the early release of the financial audit due to making the reputations of the firms in the market for attracting the investors from the globe. |
| *H5* | Ownership Structure in China-listed firms has a significant moderating impact between corporate governance attributes and financial reporting quality | Partially accepted | From the ownership structure family ownership in the Asian economies especially in China has a significant moderating impact that enhances the relationship between the corporate governance and reporting quality of financials. All the variables' relationship increases with great intentions, indicating the family interference makes the organizations that leads to decrease lags and improve the reporting quality of financials. Moreover, Ownership concentration has not significant moderating impact due to the demographic and cultural changes from other Asian economies. |

companies with qualified opinions in China have longer audit report lags compared to companies with unqualified opinions. This is likely due to the additional time and effort required to investigate and verify the misstatements that led to the qualified opinion.

The existence of an audit committee reduces the lag; it means that the Audit Committee has complete control of the accounts keeping and has great internal control that increases the financial reporting quality in China. It is crucial to acknowledge that the effectiveness of an audit committee relies on various factors, including the independence and qualifications of its members, the frequency and quality of meetings, and the level of support from top management. Hence, having an audit committee alone does not ensure improved financial reporting quality or a reduced audit report lag in China or any other country. In this way, our findings are similar to the findings of [65] who found that the audit committee increases the financial reporting quality by checking from time to time internally.

Board diligence has a significant and positive impact on the lags. It means that the large number of board meeting increase the lag and reduce the account quality in China due to the wastage of time. However, it is important for the board of directors to be diligent in their oversight responsibilities, the frequency of board meetings does not necessarily have a direct impact on the audit report lag or financial reporting quality. Moreover, the quality of board

oversight is more important than the quantity of meetings. A board that is knowledgeable about the company's operations, risks, and financial reporting processes, and is actively engaged in overseeing those areas, is more likely to contribute to higher financial reporting quality and a shorter audit report lag. In this context our findings are consistent with the findings of [66] found that board independence and expertise are positively associated with financial reporting quality in China. Moreover, from the agency's perspective, all the conflicts are resolved timely, and the decision will be acute that releasing the accounts early increase the financial reporting quality and findings consistent with the agency theory perspective.

The independent board decreased the lag because our result is significant and negative, which means that when the lag is decreased then financial reporting quality improves. Board size and CEO have the burden for the firms because our findings show that the result is significant and positive, which means that both increase the lag and decrease the reporting quality in China. Regarding the impact of an independent board on the audit report lag, there is some evidence to suggest that an independent board can improve financial reporting quality and reduce the audit report lag in China. In this context, our findings are similar to those [66] that found board independence is positively associated with financial reporting quality, as measured by the likelihood of financial restatements. [65] also found that companies with a higher proportion of independent directors have a shorter audit report lag in China. Regarding board size and CEO characteristics, the evidence is more mixed. Some studies have found a positive association between board size and the audit report lag in China [66], while others have found no significant relationship [65]. Similarly, some studies have found a negative association between CEO duality (when the CEO is also the chairman of the board and financial reporting quality in China, while others have found no significant relationship [65]. Moreover, the relationships between board characteristics and financial reporting quality or the audit report lag may depend on several factors, such as the industry, company size, ownership structure, and regulatory environment. Therefore, it is crucial to take into account the unique context and circumstances when assessing the influence of board characteristics on financial reporting quality and the audit report lag, whether in China or any other country. Family ownership has been found to have a significant positive impact on financial reporting quality, resulting in a shorter audit report lag. This could be due to the fact that family members have a vested interest in the company's operations, which may lead to an increased focus on improving the accuracy and reliability of the firm's financial reporting. Moreover, profitability also decreases the lag and increases the financial reporting quality of accounts because companies have no need for window-dressing and reporting the original words in their accounts timely. Moreover, similar to the agency theory perspective, effective corporate governance mechanisms play a critical role in mitigating agency conflicts. [18] noted that the application of agency theory within corporate governance can help to minimize such conflicts. [19] highlighted that corporate governance variables, such as ownership structure, board characteristics, and audit-related variables, can be instrumental in reducing agency problems within an organization. [20] further argued that good governance practices can limit agency conflicts and ensure the interests of shareholders are upheld. However, financial reporting accuracy is the responsibility of the management to attract the investors, the code of the corporate governance makes the financial reporting quality better. On that ground similar with the idea of stewardship theory that emphasizes the importance of providing accurate and reliable financial information to stakeholders.

In Table 7, our results find that no moderating impact is found between corporate governance's different attributes and ownership concentration. There are different reasons behind it due to the cultural, social, and other attributes effect. In Table 8, our results show that family ownership has a strong moderating impact between corporate governance attributes and

financial reporting quality. In Asian economies, mostly family-owned firms performed well due to the dual internal control and pressure by the family on the corporation that reduces the lags and increases the financial reporting quality. Family-owned firms often have dual internal control mechanisms, with the family serving as the controlling shareholder and also holding key management positions. This can create a stronger sense of ownership and accountability, which can result in greater attention to financial reporting quality and a shorter audit report lag. Additionally, family owners may have a long-term perspective on the business and may be less likely to engage in short-term financial engineering or earnings management, which can improve financial reporting quality. However, it is worth noting that family ownership can also have some drawbacks. The family owners may prioritize their interests over the interests of minority shareholders, which can lead to agency problems and weaker corporate governance. Additionally, family owners may lack the expertise or experience to effectively oversee complex financial reporting processes. However, family ownership can have a moderating impact on the relationship between corporate governance attributes and financial reporting quality, the advantages and disadvantages of family ownership should be carefully considered in the context of each individual firm and market. Research studies have shown that family-owned firms in Asian economies have performed well in terms of financial reporting quality and audit report lag, but these firms also face unique challenges related to corporate governance and minority shareholder protection.

Our results are consistent with the findings of [67] conducted a study in Pakistan that examined the impact of family ownership on financial reporting quality and revealed a positive correlation. The researchers attributed this to the presence of robust internal control and governance mechanisms in family-owned firms. Our results are also similar to the findings of [68] who investigated the impact of family ownership on audit report lag in South Korean firms and found that family ownership reduces audit report lag, which they attributed to the closer monitoring and oversight of financial reporting by family owners. Our study is also in line with [68] which conducted a study on the impact of family ownership on financial reporting quality in China and revealed that family ownership has a positive influence on financial reporting quality. The researchers attributed this to the long-term perspective and greater accountability of family owners. Our findings differ from those of a recent study conducted by [69] which examined the relationship between family ownership and financial reporting quality in Jordanian firms. They found that family ownership has a negative impact on financial reporting quality, which they attributed to weaker corporate governance and a lack of transparency in family-owned firms. Similarly, our findings contrast with the results of a study conducted by [70] which investigated the impact of family ownership on financial reporting quality in Chinese firms. They found that family ownership has a negative impact on financial reporting quality due to a higher likelihood of earnings management and weaker external monitoring in family-owned firms. However, our findings align with a study conducted by [71] which explored the impact of family ownership on financial reporting quality in Thailand. They found that family ownership has a positive effect on financial reporting quality, which they attributed to stronger family values and a sense of responsibility among family owners. Furthermore, we can conclude that family ownership can positively impact financial reporting quality and audit report lag in Asian firms due to unique advantages such as stronger internal control and governance mechanisms, closer monitoring and oversight of financial reporting, and a long-term perspective on the business. Nonetheless, it is important to note that the specific impact of family ownership may vary depending on the market and context.

## Conclusion, recommendations, and future research suggestions

Our study's main objective is to explore the relationship between corporate governance mechanism and financial report quality with moderating role of the ownership pattern. From our findings, we concluded that corporate governance's different attributes such as Auditor brand name, Existence of an audit committee, independent board, family ownership, and profitability decreases the lags and increases the financial reporting quality in China listed firms on the ground of the strong corporate governance practices followed by the Chinees organizations and also shows the effectiveness of the policy makers on the code of corporate governance in China. The code of corporate governance has different consequences that vary across countries around the globe.

Moreover, we also concluded that Auditor opinion, Board diligence Board size, and CEO duality increases the lags and decreases the financial reporting quality of China-listed firms. Furthermore, our findings show that ownership concentration has no moderating effect between corporate governance, different attributes, and financial reporting quality. Family ownership, on the other hand, has a strong moderating effect between corporate governance characteristics and financial reporting quality. From our findings, we concluded that the family firms have a significant role in following the country code of corporate governance that affects the reporting quality.

Similarly, this study advances the development and refinement of established theories like agency theory, stewardship theory, and signaling theory. By exploring the connections between corporate governance mechanisms and financial reporting quality, the study identifies and elucidate the underlying processes and mechanisms that drive these relationships in various industrial and cross-cultural settings worldwide. Moreover, this study provides empirical evidence that demonstrates the impact of corporate governance on financial reporting quality. This evidence either supports or refutes existing theories and hypotheses. However, this study provides significant policy implications for regulators, standard-setters, and other stakeholders. By identifying the mechanisms and factors that influence financial reporting quality, this study helps to inform and guide policy decisions related to corporate governance, accounting standards, and financial reporting practices. Moreover, this study suggested that strong corporate governance is essential for ensuring the integrity of financial reporting and protecting the interests of investors and other stakeholders. Companies that prioritize corporate governance, and investors and regulators that promote and enforce strong governance standards, are likely to benefit from higher financial reporting quality and lower risk of financial misreporting.

Therefore, this study also provides significant practical implications for organization management to improve their financials by focus on strengthening their corporate governance mechanisms, such as the independence of the board, the existence of audit committees, and the quality of internal controls. By doing so, companies can improve their financial reporting quality, reduce the risk of financial misreporting, and enhance the credibility of their financial statements. Investors should pay close attention to the corporate governance practices of the companies they invest in, as strong corporate governance can be an indicator of higher financial reporting quality and lower risk of financial misreporting. Investors should also use their influence to advocate for better corporate governance practices in the companies they invest in. Regulators should continue to promote and enforce strong corporate governance standards, as they play a critical role in ensuring the integrity of financial reporting and protecting the interests of investors and other stakeholders. Regulators should also encourage companies to adopt best practices in corporate governance and provide guidance and support to help companies improve their governance mechanisms. Other stakeholders, such as auditors, rating

agencies, and analysts, should take into account the impact of corporate governance on financial reporting quality when evaluating companies. Companies, investors, and regulators should take into account the specific context and market conditions of the companies they are assessing and adjust their evaluations accordingly.

Corporate governance practices may vary across the world and therefore this study has some limitations and provides future research suggestions on the corporate governance mechanism across different cultures and characteristics. Moreover, environmental factors such as pandemics and war may also affect the reporting quality due to policy uncertainty. Therefore, environmental factors and policy uncertainty should be analyzed with the code of the corporate governance mechanism in future research topics. Moreover, due to missing availability and reliability of the data, this study explains some limitations and provide the future research suggestion on how different ownership structures, such as state ownership, foreign ownership, and institutional ownership, influence reporting quality and corporate governance practices.

## Supporting information

**S1 Checklist. Author formatting checklist.**
(DOCX)

**S1 Data.**
(XLSX)

## Author Contributions

**Project administration:** Han Sun.

**Writing – original draft:** Han Sun.

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
