## [Decision Letter · Decision Letter 0]

22 May 2023

PONE-D-23-11373Corporate governance and reporting quality of accounts in China listed firms. A moderating role of Ownership pattern.PLOS ONE

Dear Dr. Sun,

Thank you for submitting your manuscript to PLOS ONE. After careful consideration, we feel that it has merit but does not fully meet PLOS ONE’s publication criteria as it currently stands. Therefore, we invite you to submit a revised version of the manuscript that addresses the points raised during the review process. Please submit your revised manuscript by Jul 06 2023 11:59PM. f you will need more time than this to complete your revisions, please reply to this message or contact the journal office at plosone@plos.org. Please include the following items when submitting your revised manuscript:A rebuttal letter that responds to each point raised by the academic editor and reviewer(s). You should upload this letter as a separate file labeled 'Response to Reviewers'.A marked-up copy of your manuscript that highlights changes made to the original version. You should upload this as a separate file labeled 'Revised Manuscript with Track Changes'.An unmarked version of your revised paper without tracked changes. You should upload this as a separate file labeled 'Manuscript'.

We look forward to receiving your revised manuscript.

Kind regards,

Wajid Khan

Academic Editor

PLOS ONE

Journal Requirements:

"This paper is a general project of 2021 Henan Higher Education Teaching Reform Research and Practice Project (Research and Practice of "Integration of Competition and Teaching" in Accounting major under the background of National Vocational College Skills Competition 2021SJGLX826)"

"This paper is a general project of 2021 Henan Higher Education Teaching Reform Research and Practice Project (Research and Practice of "Integration of Competition and Teaching" in Accounting major under the background of National Vocational College Skills Competition

2021SJGLX826)"

"This paper is a general project of 2021 Henan Higher Education Teaching Reform Research and Practice Project (Research and Practice of "Integration of Competition and Teaching" in Accounting major under the background of National Vocational College Skills Competition 2021SJGLX826)"

5. Thank you for stating the following in your Competing Interests section: "NO authors have competing interests"

7. PLOS requires an ORCID iD for the corresponding author in Editorial Manager on papers submitted after December 6th, 2016. Please ensure that you have an ORCID iD and that it is validated in Editorial Manager. To do this, go to ‘Update my Information’ (in the upper left-hand corner of the main menu), and click on the Fetch/Validate link next to the ORCID field. This will take you to the ORCID site and allow you to create a new iD or authenticate a pre-existing iD in Editorial Manager. Please see the following video for instructions on linking an ORCID iD to your Editorial Manager account: https://www.youtube.com/watch?v=_xcclfuvtxQ

8. Please ensure that you refer to Figure 3.1 in your text as, if accepted, production will need this reference to link the reader to the figure.

9. We note you have included a table to which you do not refer in the text of your manuscript. Please ensure that you refer to Tables 3.1 and 4.3 to 4.5, in your text; if accepted, production will need this reference to link the reader to the Table.

Reviewers' comments:

Reviewer's Responses to Questions

**Comments to the Author**

1. Is the manuscript technically sound, and do the data support the conclusions?

Reviewer #1: Partly

Reviewer #2: No

Reviewer #3: Yes

2. Has the statistical analysis been performed appropriately and rigorously? 

Reviewer #1: Yes

Reviewer #2: No

Reviewer #3: Yes

3. Have the authors made all data underlying the findings in their manuscript fully available?

Reviewer #1: No

Reviewer #2: No

Reviewer #3: No

4. Is the manuscript presented in an intelligible fashion and written in standard English?

Reviewer #1: Yes

Reviewer #2: No

Reviewer #3: Yes

5. Review Comments to the Author

Reviewer #1: Dear authors,

Thabnk you so much for your effort.

The manuscript was very good prepared and designed. However, I have one main concern which is about the hypotheses.

You have presented 14 hypotheses but i did not see the tests of the same 14 hypotheses if they are accepted or rejected. The authors should add one main table includes all the hypotheses and why each hypothesis statistically accepted or rejected.

Al the best.

Reviewer #2: I have reviewed the article and I am of the view that the article needs lot of improvement. The following points need consideration:

1. The research gap is not clear.

2. The reason that why China is considered research ground is missing.

3. Grounding of hypotheses is not appropriate.

4. The authors has discussed the agency theory and steward theory. These theories need to be embedded in hypotheses development and results.

5. Variable measurement is not appropriate. The board independence should be measured using proportion of independent directors.

6. None of the correlation coefficient is significant.

7. The results are not discussed in the light of literature

Reviewer #3: The current work is new topic and cater ongoing issues among the academics and policy makers. However, I have following suggestions, which can further improve the quality of the manuscript.

1. There is large number of Hypothesis….. 14 hypothesis mentioned ….I would suggest to club some the hypothesis as possible……

2. The authors used non-financial firms for the study sample…. I would suggest add some justification.. why you exclude the financial sector of the economy.

3. Page 17, “China on the basis of random sampling that has the greater market capitalization from the 1000 top market capitalized firms” mentioned by author with reference…… I am unable to understand about this…. I would suggest to add more explanation.

4. The explanation/ Definition of the depended variable i.e. Auditor lag is missing….. Throughout the paper… Also missing in the Table 3.1.

5. The correlation matrix fail to discuss the significance of the correlation coefficient among the variables. My suggestion to mentioned the significance level with correlation coefficients.

6. I would suggest to club the Table 4.3 and Table 4.4 into one table…. Which, may improve the quality and reduced the length of the paper…. Further, It looks that the current manuscript is extracted from Research thesis/project. Usually, the Table 4.1, 4.3… 2.1 not used research articles. Suggest to revisit the table numbering…….

6. PLOS authors have the option to publish the peer review history of their article (what does this mean?). If published, this will include your full peer review and any attached files.

Reviewer #1: No

Reviewer #2: **Yes: **Mr. Ali Amin

Reviewer #3: **Yes: **shahab ud din

---

## [Author Response · Author response to Decision Letter 0]

26 Jul 2023

All the suggestion are answered in the file attached

---

## [Decision Letter · Decision Letter 1]

4 Oct 2023

PONE-D-23-11373R1Corporate governance and reporting quality of accounts in China listed firms. A moderating role of Ownership pattern.PLOS ONE

Dear Dr. Sun,

Thank you for submitting your manuscript to PLOS ONE. After careful consideration, we feel that it has merit but does not fully meet PLOS ONE’s publication criteria as it currently stands. Therefore, we invite you to submit a revised version of the manuscript that addresses the points raised during the review process.

We look forward to receiving your revised manuscript.

Kind regards,

Wajid Khan

Academic Editor

PLOS ONE

Journal Requirements:

Additional Editor Comments:

I have reviewed the article entitled “Corporate governance and reporting quality of accounts in China listed firms. A moderating role of Ownership pattern”. This is an interesting study and the authors have collected a unique dataset for a unique and progressive methodology. The paper is generally well written and structured. However, in my opinion the paper has some shortcomings in regards to some data and text, and I feel this unique dataset has not been utilized to its full extent. Below I have provided numerous remarks on the text:

1. In several instances, I suggest to cite more relevant and recent studies in the introduction and methodology sections.

2. The introduction should be expanded to include importance, uniqueness and contribution to the literature.

3. You need state clearly the contributions of the paper. For example, "Consequently, the current paper seeks to make the following contributions to the existing literature. First,…, Second,…., Third, …, Fourth,… and so on". The description of the contribution needs to be more forensic, needs to be more focused.

4. The authors should discuss the relevant theories in detail and relate their findings to a specific theory of Corporate governance and Ownership pattern in the developing country context

5. There are multiple misspellings throughout the manuscript that needs attention.

I recommend Minor REVISIONS for publication after the author/s addressing the above queries and suggestions.

Reviewers' comments:

Reviewer's Responses to Questions

**Comments to the Author**

1. If the authors have adequately addressed your comments raised in a previous round of review and you feel that this manuscript is now acceptable for publication, you may indicate that here to bypass the “Comments to the Author” section, enter your conflict of interest statement in the “Confidential to Editor” section, and submit your "Accept" recommendation.

Reviewer #3: All comments have been addressed

Reviewer #4: All comments have been addressed

2. Is the manuscript technically sound, and do the data support the conclusions?

Reviewer #3: Yes

Reviewer #4: Yes

3. Has the statistical analysis been performed appropriately and rigorously? 

Reviewer #3: Yes

Reviewer #4: Yes

4. Have the authors made all data underlying the findings in their manuscript fully available?

Reviewer #3: Yes

Reviewer #4: Yes

5. Is the manuscript presented in an intelligible fashion and written in standard English?

Reviewer #3: No

Reviewer #4: Yes

6. Review Comments to the Author

Reviewer #3: authors have professionally incorporated all the my manners. They have done great job.

Best of luck.

Reviewer #4: Introduction

1. The scholar has presented quite in-depth details about the Corporate governance, FRQ and a very good approach is used to address the problem.

2. Formatting, check spelling, punctuations and grammar issue.

3. Research objectives and questions are missing.

LITERATURE Review

1. Part 2 starts with a systematic approach and provides thorough literature review, but theoretical support must be more vital to justify the topic.

2. The candidate has set the several hypothesizes from H1, H2, H3, H4 and H5 .

For example H1: Board characteristics of corporate governance have significant positive relationship with reporting quality of financials.

For hypothesis testing , if the outcome is yet to be known, it is sufficient to mentions positive impact

(drop the significant word).

3. The candidate should demonstrate how the research relates to the literature. The research findings and discussion should be linked with the literature and theoretical grounds for the conducted research in a better way.

Methodology

1. Appropriate research methods have been used. The research methodology is acceptable to justify the result.

2. Mention the proper approach of moderating technique.

Analytical tool

i. Why descriptive statistic important for your data?

ii. In this analysis part , researcher need t explain that how and why your selected sampling needs descriptive analysis and correlation analysis.

iii. Mention recent studies for ownership structure as moderating variable.

Conclusion

After all of the analysis is completed , each objective should be highlighted again in the conclusion chapter by providing discourse demonstrating where the analysis is leading the work and what the outcome is, and then linking that to literature reviews.

6 References Please check the references again and references are not as per APA format that requires attention.

7. PLOS authors have the option to publish the peer review history of their article (what does this mean?). If published, this will include your full peer review and any attached files.

Reviewer #3: No

Reviewer #4: No

---

## [Author Response · Author response to Decision Letter 1]

6 Oct 2023

Reviewer #4:

Introduction

1. The scholar has presented quite in-depth details about the corporate governance, FRQ and a very good approach is used to address the problem.

2. Formatting, check spelling, punctuations and grammar issue.

3. Research objectives and questions are missing.

Response to section Introduction:

The Grammarly, misspelling and punctuation error removed. The research objective and questions included in the introduction. Please find page 3-4 yellow highlighted as

“The main objective of this research is to assess the impact of organizational characteristics and corporate governance practices on the accuracy of financial reporting for companies listed in China. The study will specifically examine how ownership concentration and inter-organizational corporate governance (including the board of directors, audit committee, and CEO duality) are related to the reliability of financial reporting. Along with the primary goal, the study has several other specific objectives. Firstly, it aims to evaluate the current state of financial reporting quality in China. Secondly, …………………. Third, …………………. The quality of financial reports will be measured by analyzing the audit reports from the annual financial reports of listed firms.”

LITERATURE Review

1. Part 2 starts with a systematic approach and provides thorough literature review, but theoretical support must be more vital to justify the topic.

2. The candidate has set the several hypothesizes from H1, H2, H3, H4 and H5 .

For example H1: Board characteristics of corporate governance have significant positive relationship with reporting quality of financials.

For hypothesis testing , if the outcome is yet to be known, it is sufficient to mentions positive impact

(drop the significant word).

3. The candidate should demonstrate how the research relates to literature. The research findings and discussion should be linked with the literature and theoretical grounds for the conducted research in a better way.

Response to section Literature Review:

The theoretical support was added in the literature review section. Moreover, the significant words removed from the hypothesis. Please find page 8-11 yellow highlighted. The findings are related to literature and theory in the discussion in detail (please find page 23-27 and yellow highlighted).

Methodology

1. Appropriate research methods have been used. The research methodology is acceptable to justify the result.

2. Mention the proper approach of moderating technique.

Analytical tool

i. Why descriptive statistic important for your data?

ii. In this analysis part , researcher need t explain that how and why your selected sampling needs descriptive analysis and correlation analysis.

iii. Mention recent studies for ownership structure as moderating variable.

Response to section Methodology:

The moderating techniques used in the analysis are defined in the methodological section also. The data description and correlation analysis reasons are written (please find page 15 and yellow highlighted). Moreover, the recent studies on moderating variables are also included (please find page 10 and yellow highlighted).

Conclusion

After all of the analysis is completed , each objective should be highlighted again in the conclusion chapter by providing discourse demonstrating where the analysis is leading the work and what the outcome is, and then linking that to literature reviews.

6 References Please check the references again and references are not as per APA format that requires attention.

Response to section Conclusion:

The conclusion is developed according to the reviewer suggestions by supporting previous theory and literature evidence in conclusion section that justified the findings and make significant contribution of the study. Please find page 23-26 yellow highlighted. All the references are corrected and style APA numbered. 

Thank you.

---

## [Decision Letter · Decision Letter 2]

2 Nov 2023

PONE-D-23-11373R2Corporate governance and reporting quality of accounts in China listed firms. A moderating role of Ownership pattern.PLOS ONE

Dear Dr. Sun, 

Thank you for submitting your manuscript to PLOS ONE. After careful consideration, we feel that it has merit but does not fully meet PLOS ONE’s publication criteria as it currently stands. Therefore, we invite you to submit a revised version of the manuscript that addresses the points raised during the review process.

We look forward to receiving your revised manuscript.

Kind regards,

Wajid Khan

Academic Editor

PLOS ONE

Journal Requirements:

Reviewers' comments:

Reviewer's Responses to Questions

**Comments to the Author**

1. If the authors have adequately addressed your comments raised in a previous round of review and you feel that this manuscript is now acceptable for publication, you may indicate that here to bypass the “Comments to the Author” section, enter your conflict of interest statement in the “Confidential to Editor” section, and submit your "Accept" recommendation.

Reviewer #4: All comments have been addressed

2. Is the manuscript technically sound, and do the data support the conclusions?

Reviewer #4: Yes

3. Has the statistical analysis been performed appropriately and rigorously? 

Reviewer #4: Yes

4. Have the authors made all data underlying the findings in their manuscript fully available?

Reviewer #4: Yes

5. Is the manuscript presented in an intelligible fashion and written in standard English?

Reviewer #4: Yes

6. Review Comments to the Author

Reviewer #4: Comments to the Author

This work investigates “Corporate governance and reporting quality of accounts in China listed firms. A moderating role of Ownership pattern.” I am reasonably familiar with the literature on the topic and have ongoing research interest in this subject matter. I believe the topic of the research is quite interesting as the current draft attempts to unfold some hidden layers which are yet not been tackled by the previous scholars. Below, I list and comment on the findings presented:

1. The abstract should be short,” and these information: "What does the author(s) want to know? (Purpose): Why do they want to know? (gap) How do they want to know? (Methodology): What are the outcomes? (Results): What are the developments of the study? (Contribution): What are the limitations of the study? (future research direction)” must be included in the abstract. The rest of the irrelevant information should be deleted.

2. I believe the introduction is quite comprehensive. Anyway, the following information should be added at the second last paragraph in the introduction section: (a) The introduction should underline the aim of the paper; (b) stress why this aim is important (with the support of the literature); (c) summarize the methodology; and (d) describe the development of the study.

3. In my opinion, this section needs to be developed a bit. The authors have not critically reviewed the existing works.

4. The methodology should be explained properly so that the readers can understand it easily. Operational definition of variables must be presented in appropriately. Data sources must be defined. Sampling technique used must be justified.

5. Discussion should include comparing and contrasting with previous studies, discussion of the results, theoretical contribution, practical implication, and future research direction. Results should include diagnostic statistics presented in appropriate manner.

6. The conclusion should be separated from the discussion. Policy implications should be provided in light of findings of study.

Good Luck

7. PLOS authors have the option to publish the peer review history of their article (what does this mean?). If published, this will include your full peer review and any attached files.

Reviewer #4: No

---

## [Author Response · Author response to Decision Letter 2]

3 Nov 2023

Thank you for your review sir, Please find the attached Reponses

1. The abstract should be short,” and these information: "What does the author(s) want to know? (Purpose): Why do they want to know? (gap) How do they want to know? (Methodology): What are the outcomes? (Results): What are the developments of the study? (Contribution): What are the limitations of the study? (future research direction)” must be included in the abstract. The rest of the irrelevant information should be deleted.

Response:

The abstract makes the short and attractive according to the suggestion of the reviewer and removed the unnecessary information as,

“Financial reporting quality is critical for businesses, stakeholders, and government to ensure transparency and accountability. The purpose of this paper is to investigate the relationship between corporate governance, financial reporting quality, and ownership structure as a moderating factor for Chinese stock exchange-listed firms. Quantitative data of 550 listed firms from 2012 to 2022 are collected from the annual reports. For investigating the relationship between variables, panel data analysis with random and fixed effect models is used. Our results show that corporate governance's different attributes such as Auditor brand name, Existence of an audit committee, independent board, family ownership, and profitability have a significant negative impact on the audit report lag that decreases the lags and increases the financial reporting quality in China listed firms. Auditor opinion, Board diligence Board size, and CEO duality have a significant positive impact on the audit report lag that increases the lags and decreases the financial reporting quality of China-listed firms. Furthermore, our findings show that ownership concentration has no moderating effect between corporate governance, different attributes, and financial reporting quality. Family ownership, on the other hand, has a strong moderating effect between corporate governance characteristics and financial reporting quality. However, due to limitations, this study provides the opportunity for future research on corporate governance mechanisms in different cultures and environments. Moreover, this study has some significant implications for investors, policymakers, and government.”

2. I believe the introduction is quite comprehensive. Anyway, the following information should be added at the second last paragraph in the introduction section: (a) The introduction should underline the aim of the paper; (b) stress why this aim is important (with the support of the literature); (c) summarize the methodology; and (d) describe the development of the study.

Response:

The aim of the study included in the introduction section. The significance of the aims of the study with support to previous literature are written according to the suggestion (please refer yellow highlighted in the Introduction section). The methodology and development of the study is also written in Introduction according to reviewer suggestions (please refer yellow highlighted in Introduction section).

3. In my opinion, this section needs to be developed a bit. The authors have not critically reviewed the existing works.

Response:

The most relevant critical literature review added. (please refer yellow highlighted in Literature review section)

4. The methodology should be explained properly so that the readers can understand it easily. Operational definition of variables must be presented appropriately. Data sources must be defined. Sampling technique used must be justified.

Response:

The operational definitions section included references. 

“The audit report lag is the number of the days delay the audit report by the auditor to complete their audit report due to the numbers of the errors and non-compliance with the code of the corporate governance mechanism [47]. The total member in the board composition is an essential part of the corporate governance mechanism. The board size represents the number of the directors in making board decisions and taking necessary actions [48]. The independent board represents when there is independent directors involve in the board composition [48]. The CEO duality means that when CEO has the double role as CEO and the Chairman of the board of directors [50]. The numbers of the meeting of the board of directors held during the financial year is the board diligence [53]. ………………………………………………………………………………………………………… The family ownership means the shareholding by one member or family has 50% or more in the respect organization [61]. (page 13 Methodology section)

Data sources is properly defined. Sampling techniques is justifying as,

“In this study, we collected secondary data from the years 2012-2022 from the annual reports of the firms that are listed in China stock markets. The annual reports of selected firms are downloaded from the Shanghai Stock Exchange (SSE) data portal”. (page 11 Methodology section)

“A total of 550 non-financial firms are selected as a sample from the total non-financial sector listed in China. There are 4697 listed firms belonging to the non-financial sector of China that represent our populations for the study. The Sample of 550 firms is collected through random sampling from non-financial sectors of the China-listed firms [21, 45, 46]. (page 11 of methodology)

5. Discussion should include comparing and contrasting with previous studies, discussion of the results, theoretical contribution, practical implication, and future research direction. Results should include diagnostic statistics presented in appropriate manner.

Response:

The discussion is written according to the suggestion. The yellow highlighted shows the track changes in the discussion section. The findings of the study justified through previous literature in yellow highlighted.

6. The conclusion should be separated from the discussion. Policy implications should be provided in light of the findings of the study.

Response:

The conclusion is separated from the discussion. Policy implication, contribution, limitations and future research suggestions are properly written. The yellow highlighted shows the track changes in the Conclusion section.

Thank you

---

## [Decision Letter · Decision Letter 3]

20 Nov 2023

Corporate governance and reporting quality of accounts in China listed firms. A moderating role of Ownership pattern.

PONE-D-23-11373R3

Dear Dr. Han Sun,

We’re pleased to inform you that your manuscript has been judged scientifically suitable for publication and will be formally accepted for publication once it meets all outstanding technical requirements.

Kind regards,

Wajid Khan

Academic Editor

PLOS ONE

Additional Editor Comments (optional):

Reviewers' comments:

Reviewer's Responses to Questions

**Comments to the Author**

1. If the authors have adequately addressed your comments raised in a previous round of review and you feel that this manuscript is now acceptable for publication, you may indicate that here to bypass the “Comments to the Author” section, enter your conflict of interest statement in the “Confidential to Editor” section, and submit your "Accept" recommendation.

Reviewer #4: All comments have been addressed

2. Is the manuscript technically sound, and do the data support the conclusions?

Reviewer #4: Yes

3. Has the statistical analysis been performed appropriately and rigorously? 

Reviewer #4: Yes

4. Have the authors made all data underlying the findings in their manuscript fully available?

Reviewer #4: Yes

5. Is the manuscript presented in an intelligible fashion and written in standard English?

Reviewer #4: Yes

6. Review Comments to the Author

Reviewer #4: All comments have been incorporated

A proof reading is required before making final submission

All reporting of results must follow APA style

7. PLOS authors have the option to publish the peer review history of their article (what does this mean?). If published, this will include your full peer review and any attached files.

Reviewer #4: No

---

## [Editor Report · Acceptance letter]

22 Nov 2023

PONE-D-23-11373R3 

Corporate governance and reporting quality of accounts in China-listed firms. A moderating role of Ownership pattern. 

Dear Dr. Sun:

I'm pleased to inform you that your manuscript has been deemed suitable for publication in PLOS ONE. Congratulations! Your manuscript is now with our production department. 

Kind regards, 

on behalf of

Dr. Wajid Khan 

Academic Editor

PLOS ONE